# Diet Quality and Upper Gastrointestinal Cancers Risk: A Meta-Analysis and Critical Assessment of Evidence Quality

**DOI:** 10.3390/nu12061863

**Published:** 2020-06-23

**Authors:** Sara Moazzen, Kimberley W. J. van der Sloot, Roel J. Vonk, Geertruida H. de Bock, Behrooz Z. Alizadeh

**Affiliations:** 1Department of Epidemiology, University Medical Center Groningen, University of Groningen, 9712 CP Groningen, The Netherlands; k.w.j.van.der.sloot@umcg.nl (K.W.J.v.d.S.); g.h.de.bock@umcg.nl (G.H.d.B.); b.z.alizadeh@umcg.nl (B.Z.A.); 2Department of Celbiology, University Medical Center Groningen, University of Groningen, 9712 CP Groningen, The Netherlands; roelvonk@gmail.com

**Keywords:** healthy diet, gastrointestinal neoplasms, stomach neoplasms, esophageal neoplasm

## Abstract

We aimed to assess the effect of a high-quality diet on the risk of upper gastrointestinal cancer and to evaluate the overall quality of our findings by searching PubMed, EMBASE, Web of Science, Cochrane, and the references of related articles to February 2020. Two reviewers independently retrieved the data and performed the quality assessments. We defined the highest-quality diet as that with the lowest Diet Inflammatory Index category and the highest Mediterranean Diet Score category. Overall odds ratios and 95% confidence intervals were estimated for upper gastrointestinal cancer risk comparing the highest- versus lowest-diet quality. A random-effects meta-analysis was then applied with Review Manager, and the quality of the overall findings was evaluated with the Grading of Recommendations Assessment, Development, and Evaluation approach. The highest-quality diets were significantly associated with reduced risk of upper gastrointestinal cancers, achieving odds ratios of 0.59 (95% confidence interval: 0.48–0.72) for the Diet Inflammatory Index, pooling the findings from nine studies, and 0.72 (95% confidence interval: 0.61–0.88) for the Mediterranean Diet Score, pooling the findings from 11 studies. We observed a minimum of 69% heterogeneity in the pooled results. The pooled results were graded as low quality of evidence. Although it may be possible to offer evidence-based general dietary advice for the prevention of upper gastrointestinal cancers, the evidence is currently of insufficient quality to develop dietary recommendations.

## 1. Introduction

Upper gastrointestinal (UGI) cancers, including those of the stomach, esophagus, and nasopharynx, are a leading cause of morbidity and mortality worldwide, accounting for some 1.6 million new cases and 1.3 million deaths each year [1]. Both unmodifiable and modifiable factors are associated with their development, and dietary components have received particular attention as a potentially modifiable factor. Given the proven effect of distinct nutritional components in various chronic diseases, several dietary indices have been developed to estimate diet quality in different populations. However, these indices differ in terms of the basis on which they are scored and the dietary components that they cover.

The Diet Inflammatory Index (DII), the Mediterranean Diet Score (MDS), and the Healthy Eating Index (HEI) are among the most well-characterized diet indices (Appendix A) [2,3,4]. To date, research seeking to quantify the risk of UGI cancer using these indices has yielded inconclusive results. Several observational studies have reported no effect on the incidence of gastric cancer when implementing a high-quality diet according to DII [5] or MDS [5,6,7] criteria, or on the risk for esophageal cancer, according to MDS criteria [6]. However, findings from a prospective study demonstrated approximate reductions of 40% in the risk of gastric cancer [8] and 50% in the risk of esophageal cancer [9], when consuming a high-quality diet defined by the DII. Likewise, others have reported beneficial effects on the risk of UGI cancer when consuming high-quality diets, as measured by the MDS [10,11] and HEI [12].

Variations in the methods, such as the study design, the geographic region, the study population, the method of quantifying diet quality, and the anatomical tumor site are considered to account for the existing discrepancies, resulting in inconsistency in meta-analyses. This problem has been compounded by the lack of analysis of the overall quality of existing data, and together, these have prevented the development of meaningful dietary recommendations for the prevention of UGI cancer [13,14,15,16]. Specific issues are that researchers have pooled the results for only a single type of dietary index [13,14], that the number of included studies has been limited [14,16], and that the pooling of results has mainly been limited to gastric cancer and not to other types of UGI cancer [13,16]. Given the limitations of the existing evidence, we must not only assess the role of diet quality on UGI cancer risk but also evaluate the quality of the overall findings.

In this text, we report a systematic review and meta-analysis with two aims. First, we assessed the effect on UGI cancer risk of diet quality measured by diet indices. Second, we aimed to study whether the existing evidence on the beneficiary role of diet quality in the prevention of UGI cancer could be translated into evidence-based dietary recommendations for the prevention of UGI cancers.

## 2. Materials and Methods

### 2.1. Study Protocol and Search Strategy 

We formulated the search strategy for PubMed, Embase, Web of Science, and Cochrane with the assistance of a medical librarian (Appendix A). The strategy included all studies evaluating the role of diet quality and UGI cancer risk, published up to February 2020, with no language restrictions. In addition, the reference lists of included studies were searched to find related studies. Two investigators (SM and KWJS) completed these steps independently and resolved discrepancies with a third investigator (BZA). The review process was based on the PRISMA-P guidelines [17].

### 2.2. Eligibility Criteria, Data Extraction

We included studies measuring diet quality with a diet index as part of their a priori design. In these, UGI cancer cases were required to be identified by the presence of one or more of following: self-report questionnaire or pathology records reviewed by a trained physician, linkage to a cancer registry system, or linkage with a mortality records system.

SM and KWJS extracted the following information: study design, first author, publication year, study country, applied diet index, number of included food components, gender, and age (mean or range). The total population size and incident cases were recorded for cohort studies, and the number of cases and controls were recorded for case-control studies. They also recorded the number of adjusting variables, the outcomes, and the most adjusted risk estimates with corresponding 95% confidence intervals (CIs) comparing the highest and lowest diet index categories.

### 2.3. Study Quality Assessment

Finally, study quality of eligible studies was appraised by the Newcastle–Ottawa quality assessment scale, according to three parameters: selection, comparability, and exposure for cohort and case-control studies as the following [18].

As for cohort studies, the following items were evaluated as (i.) *Selection.* The representativeness of the exposed and unexposed populations, and the adequacy of outcome demonstration; (ii.) *comparability.* Control for age, gender, and at least three additional risk factors, including body mass index, ethnicity, family history of gastrointestinal (GI) cancers, smoking, alcohol, physical activity, dietary supplement intake, helicobacter pylori, and gastroesophageal reflux disease (GERD); (iii.) *Exposure/outcome.* The methods used for outcome assessment, the adequacy of outcome follow-up (i.e., >10 yrs. for UGI cancers), and the details of any loss to follow-up; for case-control studies, the selection and exposure parameters were modified, as follows (i.) *selection.* The adequacy and representativeness of the case definition, the control selection (e.g., community or hospital), and the confirmation of no history of GI cancer among controls; (ii.) *exposure.* Whether the same methods were applied to cases and controls for exposure assessment (e.g., secured records, validated questionnaires, or self-report assisted by a healthcare practitioner). 

Eventually, study quality was ranked as low (≤3 stars), moderate (4–6 stars), or high (≥7 stars). The maximum scores allocated for study quality were nine stars for a cohort study and eight stars for a case-control study. Any disagreements were resolved by discussion and mutual agreement.

### 2.4. Data Analysis

When there were less than three studies for a certain diet quality measurement, studies were not pooled, as in this situation, the grading of the quality of overall findings is limited. Therefore, in a quantitative meta-analysis, diet quality indices, which were investigated in three or more studies, were included. Hence, we included DII and MDS in the analyses. For the DII, we used the most adjusted risk estimates for the lowest compared to the highest category [19], while the reverse was used for the MDS. By the next step, we considered hazard ratios to be relative risks (RRs), and in turn, converted these to odds ratios (ORs), using the following formula [20]:RR = {OR/((1 − P0) + (P0 × OR))}
where, P0 is the mean incidence rate of gastric/esophageal cancer in the general population within years of the corresponding study was conducted [21]. The standard error (SE) was then calculated with the following equation:SElog (RR) = {(SElog (OR) × log(RR))/log(OR)}

The natural logarithms of the ORs, together with their SEs and corresponding CIs, were calculated later. Publication bias was assessed by visual evaluation of funnel plots and Egger’s regression tests for funnel plot asymmetry. To handle the publication bias, we applied the trim and fill methods, as proposed by Duval and Tweedie [22]. In this way, effect sizes for missing studies were estimated, and after adding these effects sizes, the pooled effect size was recalculated. The Cochran’s Q statistic, I2 Index, and P of heterogeneity (Phet) assessed homogeneity. We used random-effects models unless the I^2^ was ≤25% (indicating low heterogeneity), in which case we applied fixed-effects models. The pooled results for the DII and MDS were determined by the inverse variance method. For the sensitivity analysis, we assessed the consistency in the pooled results by excluding one study at a time and recalculating the pooled effect at each step. We also conducted stratified analyses to check the consistency of the findings within strata and to explore potential sources of heterogeneity by geographical region (Asia, Europe, and North America), gender (male and female), and tumor site (gastric and esophageal). Analyses were conducted using Review Manager, Version 5.3 (The Nordic Cochrane Centre, The Cochrane Collaboration, Copenhagen, Denmark) and Comprehensive Meta-Analysis software, version 2.2 (Biostat, Englewood, NJ, USA).

To evaluate whether the existing evidence is of sufficient quality in developing evidence-based dietary recommendations for the prevention of UGI cancers, we evaluated the quality of the pooled results on the risk of UGI cancer for both the DII and MDS, using the Grading of Recommendations Assessment, Development, and Evaluation (GRADE) approach [23]. The probable limitations, including risk of bias, inconsistency, indirectness, imprecision, and any other considerations, were defined according to these guidelines [23]. We applied the following scores: 0 = not serious, once all components met the criteria; −1 = serious when 1–2 components did not meet the criteria; and −2 = very serious, when >2 components did not meet the criteria. We later upgraded the quality of the overall findings for a large effect size, as follows: 0 = not present; +1 = the pooled effect size showed a decrease of ≤2 in the risk of UGI cancer, and +2 = the pooled effect size demonstrated a >2 times lower risk of UGI cancer.

We obtained the required information on cancer incidence rates for the country of corresponding studies within the follow-up period, using the databases offered by the website of Global Cancer Observatory. This web-based platform presents worldwide cancer statistics to provide information both for cancer research and cancer control. The data on global epidemiological profile of cancers are supplied by several projects of the International Agency for Research on Cancer, including GLOBOCAN, Cancer incidence in five continents, and cancer survival; in Africa, Asia, the Caribbean, and Central America. The required information on applied methodology in this systematic review and pooling the findings, and quality assessment of overall findings were obtained from the Cochrane handbook and website [1,20].

## 3. Results

Of the 53 studies retrieved for full-text review, 24 met the selection criteria (Figure 1 and Table 1). UGI cancers were identified by self-reported questionnaire [7,8,10], physician-reviewed pathology records [7,8,10], linkage to cancer registry system [5,6,7,8,10,11], or pathological confirmation [9,12,24,25,26,27,28,29,30,31,32,33,34,35,36,37]. All included cohort studies were rated as high-quality (7–9 points) (Appendix A).

### 3.1. UGI Cancer Risk Based on the DII

We included two cohort studies [5,8] and seven case-control studies [9,27,28,29,30,31,36] that used the DII. The quantitative analysis included 519,741 subjects from cohort studies (mean follow-up, 14.5 years) and 5776 subjects from case-control studies. Although visual assessment revealed an asymmetry in funnel plot (Appendix A), no significant publication bias was detected by Eggers regression tests (z = 1.04, *p* = 0.26).

Low DII scores did indicate statistically significant protection against UGI cancer, with a pooled OR of 0.59 (95% CI: 0.48–0.72). The heterogeneity remained significant after excluding studies with extreme findings, identified by funnel plot asymmetry [26,27] or recalculating the effect size by trim and fill method (Appendix A).

The direction of the pooled effects by gender was not consistent with the overall effect (Figure 2), and heterogeneity among the included studies within each strata was significant (Appendix A).

The pooled results were scored −2 when grading the overall evidence level. This was because the findings were limited to Europe and Asia (scoring −1) and were inconsistent (scoring −1) (Table 2 and Appendix A).

### 3.2. UGI Cancer Risk Based on the MDS

We included five cohort studies [5,6,7,10,11] and six case-control studies [12,26,32,33,35,37] that used the MDS. These cohort studies included 1,022,760 subjects (mean follow-up = 13.32 years) and the case-control studies included 10,447 subjects. Again, there was an asymmetry in the funnel plot, and significant publication bias was detected by Eggers regression tests (z = −3.74, *p* = 0.01) (Appendix A). Findings showed significant heterogeneity (I^2^ = 69% (χ^2^ = 31.81, P_het_ = 0.0004). However, after excluding studies with extreme findings [32,35], the asymmetry in the funnel plot improved, and the observed heterogeneity became non-significant. Given the observed significant publication bias, we applied the trim and fill method to calculate an unbiased effect size. Similar to the observed overall estimates, high scores in the MDS had a significant protective effect on the risk of UGI cancer, with a pooled OR (adjusted for trim and fill value) of 0.72 (0.61–0.88). Moreover, the direction of the pooled findings did not change in the sensitivity analyses, which was consistent in all stratified analyses except for geographic region, Figure 2. We detected substantial heterogeneity in the analyses stratified by study type, gender, and tumor site (Appendix A).

The quality of evidence was given a score of −3 due to the findings being restricted to specific geographic regions, such as Europe (scoring −1), the lack of consistency (scoring −1), and the possibility of publication bias (scoring −1) (Table 2 and Appendix A).

## 4. Discussion

In this systematic review and meta-analysis of 21 studies, which included a total of 1,558,724 individuals, we showed that a high-quality diet quantified by the DII and MDS was significantly associated with a reduced risk of UGI cancer. However, the data suffered a few limitations. The overall beneficial effect was not consistent for diet quality measured by the DII when stratified by gender. In addition, a high-quality diet measured by the MDS was significantly and consistently associated with a lower risk of UGI cancer when stratified by gender and tumor site. The overall quality of the evidence was graded as low regarding the beneficial effect on UGI cancer risk for a high-quality diet measured by dietary indices.

### 4.1. The DII and UGI Cancer Risk

The overall beneficial effect of a high-quality diet measured by the DII was consistent with that of a recent meta-analysis pooling the findings of three case-control studies and reporting a 2.11-times reduction in the risk of gastric cancer [13]. By contrast, Boden et al. reported that diet quality measured by the DII had no impact on UGI cancer risk in a cohort study [5]. This discrepancy could be due to several factors, including that 15 food components with anti-inflammatory effects were missing in the DII calculation, that confounders specific to UGI cancer were not adjusted for (e.g., Helicobacter pylori and GERD), and that there was reduced power in the study by Boden et al. [5], (e.g., few incident cases).

In our current meta-analysis, two studies presented extreme findings compared with the pooled results [28,29]. The latter might have resulted from the lower quality of the hospital-based case-control design, the limited number of cases, and the lack of data on at least 14 food components for DII calculation [28,29]. The recent meta-analysis by Du et al. in 2019 found consistent findings with our overall findings [16]. However, the previous meta-analysis focused solely on gastric cancer and diet indices based only three included studies. They found a beneficiary effect for high diet quality in the prevention of gastric cancer [16]. The present study, however, included a broader range of studies of upper GI cancers in the esophagus, and stomach. We included one additional study, which was carried out on gastric cancer, and five additional studies on other UGI cancers. Accordingly, the consistency observed in the earlier meta-analysis was not confirmed.

Based on the results of this meta-analysis, gender may be considered a key factor contributing to high overall levels of inconsistency. However, other factors also appeared to contribute to inconsistencies in pooled results. These included substantial inter-tumor heterogeneity in UGI cancers and adjustment for different covariates, as well as variability in the dietary component of the DII, the categories applied for analysis with the DII, and the methods used to validate the food frequency questionnaires. The low quality of the overall findings, mainly due to the observed inconsistency and indirectness (limited studies from Asia and no evidence for North America), reduced our confidence in the observed benefit of a low DII score being associated with reduced UGI cancer risk. Similarly, the diversity in the scoring system based on the inflammatory response of food components (ranging from 18 to 45 food components) and the low transparency hamper the ability to generate dietary recommendations for preventing UGI cancers based on the DII scoring system. Thus, further research is required using a consistent scoring system for the DII, especially in the less-well represented areas, if we are to draw robust conclusions about applying the related evidence to generate dietary guidelines for the prevention of UGI cancer.

### 4.2. The MDS and UGI Cancer Risk

The benefit of an MDS-based high-quality diet for preventing UGI cancer agreed with the conclusions of a recent meta-analysis that included four of the eleven studies in our meta-analysis. Despite the overall beneficial findings in the present analysis, however, we found no beneficial effect in the four included studies [5,6,7,12]. This may have been due to the diversity in UGI cancer subgroups, with these studies including all UGI cancers [5], only gastric cancer [6,7], only esophageal cancer [6], or only and nasopharyngeal cancer [12]. Inter-tumor heterogeneity, failure to adjust for confounders specific to UGI cancer (e.g., Helicobacter pylori [6,7] and GERD [6]), the limited number of incident cases [5], and the use of a single dietary assessment [6,12] may also have contributed. Interestingly, although the findings from two other studies [32,35] were similar to those of the pooled results, they showed inconsistency and publication bias. These may have arisen due to limitations inherited from the included case-control studies, as well as being hospital-based [32,35], only including cases and controls from one hospital [32], and only matching cases and controls by age [32].

The inconsistency in the findings by geographic region confirms that study region can explain much of the overall discrepancy in the results. In the current meta-analysis, our findings were mostly restricted to Europe, necessitating investigation in other geographic areas to confirm or refute whether the geographic region is responsible for the observed inconstancies. Our data do, however, support excluding gender and tumor type as causes of heterogeneity, based on the observation that benefits persisted in these strata. Thus, factors such as inter-tumoral heterogeneity and differences in the covariates adjusted for in the included studies may remain potential sources of heterogeneity. Accordingly, research must confirm the generalizability of high adherence to the MDS for preventing UGI cancer.

The low overall quality of our findings, mainly due to inconsistency and a lack of data for North America and Asia, precludes developing global dietary recommendations for the prevention of UGI cancer. Nonetheless, the high consistency in the findings of prospective research highlights the high quality of the evidence for general dietary recommendations to prevent UGI cancer. Overall, prospective investigations with broader scope are warranted to compensate for the limited evidence from underrepresented regions. In turn, it is anticipated that this will aid the development of global dietary guidelines for the prevention of UGI cancer.

### 4.3. Other Diet Quality Indices and UGI Cancers

Findings from investigations quantifying diet quality based on Chinese Healthy Food Patterns [38], Dietary Guidelines for Americans Adherence Index [24], Diet Quality Index [12], Food Diet Score [39]; Index of Nutritional Quality [40] and empirically defined food indices [32] have demonstrated the beneficiary role of high diet quality in the prevention of UGI cancers. Pooling the findings from two studies assessing diet quality based on HEI [6,12] revealed the preventive effect of high diet quality on UGI cancers. However, the two studies showed a high degree of heterogeneity. Variation in applied diet quality indices, applied solely by one or two investigations, halted pooling the effect, and assessing the quality of pooled findings were not feasible.

### 4.4. Study Strengths and Limitations

This research benefitted from an extensive assessment of the role of diet quality, measured by commonly applied dietary indices, on the risk for UGI cancer. Specifically, we applied robust grading of the overall quality of our findings. We provided a comprehensive overview of the repercussion of existing evidence for developing dietary recommendations to prevent UGI cancer. Nevertheless, the pooled findings were inevitably affected by the limitations inherited from the included observational studies, despite rating overall quality. In these observational studies, difficulties exist in a person’s habitual diet assessment. First, dietary assessment is dependent on the memory of the study subjects. Second, dietary assessments are prone to possible recall biases in unconscious over-reporting of a healthy diet and under-reporting of an unhealthy diet and fluctuations in diet habits based on changes in the environmental and lifestyle situations. This, in turn, leads to an imbalanced recall among participants.

Furthermore, diet quality assessment has mostly been done by a single dietary intake assessment, while dietary habits change over time. Accordingly, while multiple dietary assessments over time would represent more precise actual dietary habits, a single dietary assessment may fail to provide a real reflection of dietary habits. Among other limitations of this study is the substantial heterogeneity in pooled results for the DII and MDS, necessitating that we conduct extensive stratified analyses of the main confounding factors (i.e., study design, geographic region, gender, and tumor size). Another shortcoming is that the findings were mostly restricted to a specific geographic region; consequently, the results cannot be generalized. The overall quality of the findings for both indices was, therefore, downgraded.

## 5. Conclusions

Although our findings suggest a possibility to offer evidence-based general dietary advice for the prevention of UGI cancer, the evidence is currently of insufficient quality to develop dietary recommendations. Developing a dietary recommendation for UGI prevention will provide a unified approach that could be clinically applied to improve the diet quality of people at higher risk of UGI cancers, leading to better prevention, and can also be used as a mandate to improve the quality of diet in patients already diagnosed with UGI cancers which, in turn, may lead to a better prognosis.

## Figures and Tables

**Figure 1 nutrients-12-01863-f001:**
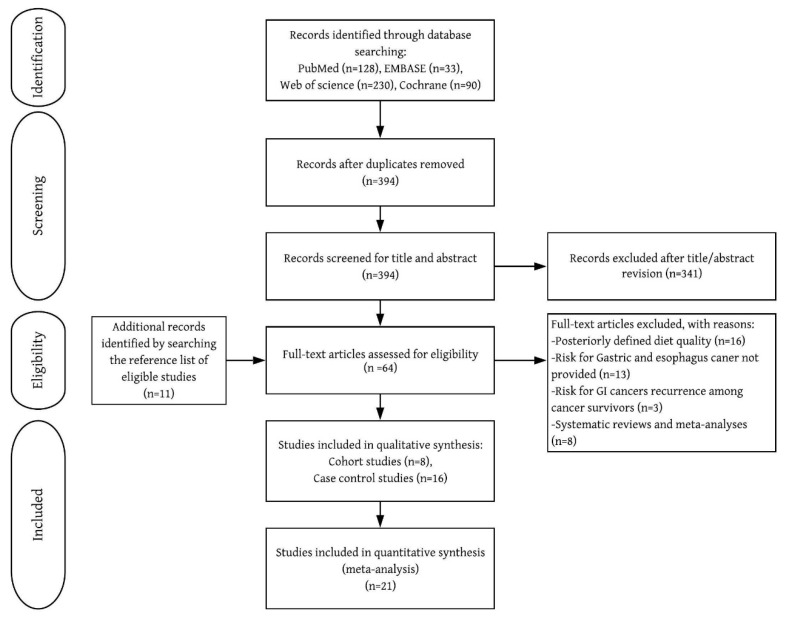
Flow chart of inclusion of relevant studies.

**Figure 2 nutrients-12-01863-f002:**
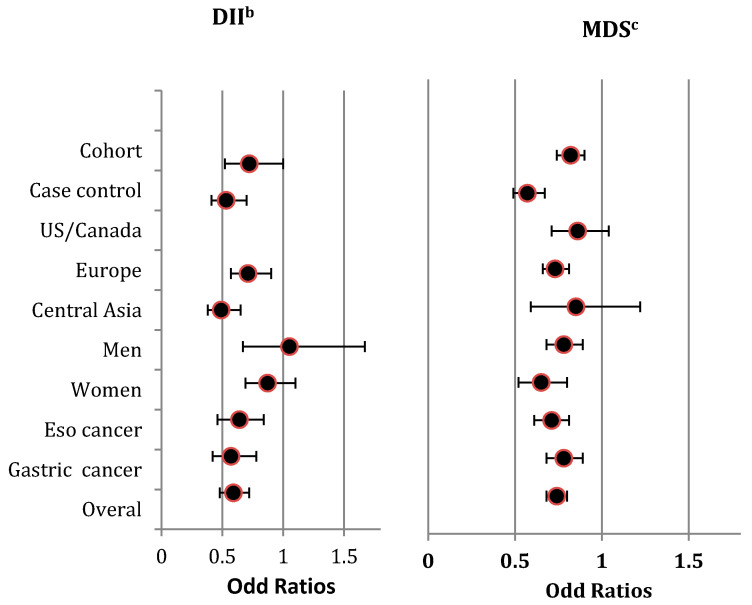
Summary risk estimates for highest diet quality compared to lowest diet quality concerning UGI cancers, stratified by study design (i.e., Cohort studies/Case-control studies) geographic region (i.e., US/Canada, Europe, Central Asia) gender (i.e., Men, Women), tumour site (i.e., Esophagus cancer, Gastric cancer) and overall estimate ^a^. ^a^ The Summary risk estimates are from pooling the reported ORs by included studies for the highest diet quality compared to the lowest, measured by DII and MDS with respect to UGI cancers. ^b^ The lowest categories of DII measure the highest diet qualities, and the highest categories of DII measure the lowest diet quality. ^c^ The highest categories of MDS measure the highest diet qualities, and the lowest categories of MDS measure the lowest diet qualities. **Abbreviations:** DII, Diet Inflammatory Index; Eso, Esophagus; MDS, Mediterranean Diet Score; OR, Odds Ratio; UGI, Upper Gastro-Intestinal; US, United States.

**Table 1 nutrients-12-01863-t001:** General characteristics of included studies in the systematic review and meta-analysis of diet quality quantified by dietary indices and risk for UGI cancers.

**Cohort Studies**								
**Author (ref), Year Country**	**Diet Index (No of Components, Categories)**	**Gender, Age (Mean or Range)**	**Population**		**No. Variable in Adjustments**	**Outcome**	**Follow Up (Years.)**	**Quality Score ***
			**Total**	**Incident Cases**				
Boden [5], 2019 Sweden	DII (30, Tertiles) MDS (8, Tertiles)	M/F 30–60	42,511	163	5	Gastric cancer	15	8
Schulpen [11], 2019 Netherlands	MDS (8, Scores 0–3 compared to 6–8)	M/F 55–69	33,655	1048	11	Gastric/Esophageal cancer	20.3	7
Agudo [8], 2018 Spain	DII (45, Quartiles)	M/F 40–60	476,160	913	8	Gastric cancer	14	9
Zhang [38], 2017 China	CHFP (11, Top three quintile compared the first two quintiles)	M 40–74	59,503	477	8	Gastric cancer	9.28	-
Buckland [7], 2015 EPIC cohort	MDS (9, scores ≤8 compared to <8)	M/F 25–70	461,550	662	7	Gastric cancer	11.4	8
Li [6], 2013 US	MDS (7, Quintile), HEI (12, Quintile)	M/F 51–70	494,968	1802	10	Gastric/Esophageal cancer	11	8
Jeumink [39], 2012 EPIC	FDS (8, Tertiles)	M/F 42–60	452,269	475	10	Gastric/Esophageal cancer	8.2	9
Buckland [10], 2010 EPIC cohort	MDS (9, Tertiles)	M/F 35–70	485,044	449	6	Gastric cancer	8.9	8
**Case-Control Studies**								
**Author (ref) Year, Country**	**Diet Index (No of Component, Categories)**	**Gender, Age (Mean or Range)**	**Population**		**No. Variable in Adjustments**	**Outcome**		**Quality Score**
			**Case**	**Control**				
Abe [9] 2018, Japan	DII (25, Quartiles)	M/F 40–70	433	1296	8	Esophagus cancer		7
Tang [36] 2018, China	DII (23, Quartiles)	M/F Control: 60.6 (11.8) Cases: 61.4 (11.0)	359	380	9	Esophagus cancer		7
Vahid [40] 2018, Iran	INQ (31, 1 point increment)	M/F 48.3 ± 10.7	82	95	7	Gastric cancer		5
Castelo [26] 2018, Spain	MDS (26, Quartiles)	M/F 63.93(11.43)	295	3040	10	Gastric cancer		8
Vahid [28] 2018, Iran	DII (31, Scores ≤−1.77 compared >−1.77)	M/F 48.3 ± 10.7	82	95	10	Gastric cancer		7
**Case-Control Studies**								
**Author (ref) Year, Country**	**Diet Index (No of Component, Categories)**	**Gender, Age (Mean or Range)**	**Population**		**No. Variable in Adjustments**	**Outcome**		**Quality Score**
			**Case**	**Control**				
Shivappa [30] 2016, Italy	DII (45, >−1.20 compared to >1.28)	M/F 39–77	304	743	11	Esophageal cancer		7
Lee [27] 2017, Korea	DII (35, Tertiles)	M/F 51.25 (9.58)	388	776	8	Gastric cancer		7
Stojanovic [32] 2017, Italy	MDS (10, Tertiles)	M/F 60–80	223	S	3	Gastric cancer		6
Wang [12], 2016, China	MDS, HEI, DQI (9, 14, 11, Quartiles)	M/F 47.4 (9.00)	600	600	16	Nasopharynx cancer		7
Shivappa [30] 2016, Italy	DII (45, Quartiles)	M (55)/F 50–70	230	547	7	Gastric cancer		7
Shivappa [29] 2016, Iran	DII (27, Score ≤1.2 compared to >1.2)	M(40)/F 58.0(10.4)	47	96	8	Esophageal cancer		7
**Case-Control Studies**								
**Author (ref) Year, Country**	**Diet Index (No of Component, Categories)**	**Gender, Age (Mean or Range)**	**Population**		**No. Variable in Adjustments**	**Outcome**		**Quality Score**
			**Case**	**Control**				
Praud [9] 2013, Italy	MDS (29, Score 0–3 compared to ≥6)	M/F 50–70	999	2628	9	Gastric cancer		7
Jessri [24], 2011, Iran	DGAI (15, Tertiles)	M/F 40–75	50	100	8	Esophageal cancer		7
Campbell [34] 2007, Canada	Food index (5, Quartiles)	M/F 20–70	1169	2332	7	Gastric cancer		8
Bosetti [35] 2003, Italy	MDS (8, Scores <3 compared to ≥6)	M/F 45–56	304	743	7	Esophageal cancer		7
Stefani [32] 2003, Uruguay	Empirically defined diet indices (19, Tertiles)	M/F 30–89	240	960	8	Gastric cancer		6

***** Quality Scores was based on New Casstel Ottawa scale. Abbreviation: CHFP, Chinese Healthy Food Patterns; DGAI, Dietary Guidelines for Americans Adherence Index; DII, Diet inflammatory index; DQI, Diet Quality Index; F, Female; FDS, Food Diet Score; HEI, Healthy Eating Index; INQ, Index of Nutritional Quality; M: Male, MDS, Mediterranean diet score; OR, Odd ratio; US, United States.

**Table 2 nutrients-12-01863-t002:** The overall quality of evidence diet quality quantified by dietary indices and risk for UGI cancers in pooled findings from eligible studies.

Summary of Findings	Certainty Assessment ^a^
No of Studies	Study Design	Risk of Bias	Inconsistency	Indirectness	Imprecision	Considerations	Relative (95%CI)	Certainty
**DII and UGI cancers (Follow up: mean 14.5 years)**	
9	Observational	Not serious	Serious ^b^	Serious ^c^	Not serious	-	OR 0.59 (0.48 to 0.72)	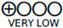 (score = −2) ^e^
**MDS and UGI cancers ( Follow up: mean 13.32 years)**	
11	Observational	Not serious	Serious ^b^	Serious ^d^	Not serious	Publication bias	OR 0.72 (0.61 to 0.88)	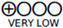 (score = −3) ^f^

^a^ The quality assessment was based on the GRADE approach. ^b^ High levels of heterogeneity in pooled findings. ^c^ Results restricted to Europe and Asia. ^d^ Findings from US/Canada and Asia limited to only one study. ^e^ The Overall certainty was downgraded due to the observational design of included studies, inconsistency and indirectness. ^f^ The Overall certainty was downgraded due to the observational design of included studies, inconsistency, indirectness, and detected publication bias. **Abbreviations**: CI, Confidence Interval; DII, Diet Inflammatory Index; GRADE, Grading of Recommendations Assessment, Development, and Evaluation; MDS, Mediterranean Diet Score; UGI, Upper Gastro-Intestinal; US, United States.

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
