# Peer review of "Diet Quality and Upper Gastrointestinal Cancers Risk: A Meta-Analysis and Critical Assessment of Evidence Quality"

_nutrients, 2020, doi:10.3390/nu12061863_

Round 1

Reviewer 1 Report

The authors aimed to evaluate the effect of diet quality in the risk of upper gastrointestinal cancer risk through a systematic review and a meta-analysis. Although the result of this study would be interesting in general, I have several concerns and suggestions on the manuscript.

  1. In lines 65-66, the authors mentioned in the manuscript that SM and KWJS independently reviewed relevant studies, and it was checked by BZA. However, in line 319 for author contributions, data curation is mentioned to be conducted by SM only. The authors may confirm this part.
  2. Line 102: The authors may specify how the incidence of UGI cancer in the general population is selected. Was the year of the original study (publication year/participant recruitment year) the same with the year of the incidence rate? In the case of the original study participants were recruited in a period of several years, how is this incidence rate adjusted or considered?
  3. Lines 120-125: Do the authors have any reasons or rationales to score the GRADE quality of the evidence?
  4. Lines 134-136: As there is the presence of publication bias, do the authors consider applying trim and fill method to handle this part?

    5. Figure 1 should be revised to explain the flow chart of selecting relevant references.

    6. Table 1: Although DII and MDS are mostly reported from individual studies, the findings of other dietary scores such as CHFP, FDS, INQ, DGAI, food index, empirically defied diet indices can be described. The authors may also specify the reason for not performing pooled analysis for HEI, although the HEI is reported from two different individual studies. The number of categories for each dietary index should be provided. The citation for individual studies has to be checked.

    7. Lines 246-248: The reference is needed.

    8. Line 332: Full information about this reference is needed.

    9. The reference format should be consistent.

  1. The words ‘gender’ and ‘sex’ should be used correctly and consistently throughout the manuscript.

Author Response

Dear Reviewer No.1.

Reviewer 2 Report

This is an ambitious, well-designed meta analyses study. The authors included data from peer reviewed articles listed in PubMed, EMBASE, Web of Science and Cochrane. Eleven more from other sources. The number of subjects covered in this study exceeded a million. However, it is not clear how many subjects were included in the European study.

               Food availability determines the dietary habits. Diet also has long term effect. Genetics plays a role too. Therefore, considering the complexity, it is not surprising that it is an inconclusive study. I would recommend restricting this meta analyses to European population, which may suggest areas to focus to obtain more conclusive results.

               If not already obtained, I recommend peer review from a qualified statistician.

Q: The study included reports published till Feb. 2020. What is the start date for this study? If the study included points from

Q: The authors mentioned they had evaluated data from 11 studies from other sources. I recommend describing these sources.

Recommendations:

I expect this meta analyses article may be read by a wider audience. Therefore, I recommend providing more details about the databases used. For example, the Cochrane and the Globocan databases have invaluable information.

Author Response

Comments from reviewer #2

We appreciate very much reviewer number 2 for reviewing and commenting on our manuscript.  Here follow we address his comments and highlight the corresponding changes we made on our manuscript.

2.1. This is an ambitious, well-designed meta analyses study. The authors included data from peer reviewed articles listed in PubMed, EMBASE, Web of Science and Cochrane. Eleven more from other sources. The number of subjects covered in this study exceeded a million. However, it is not clear how many subjects were included in the European study.

Reply: In stratified analysis, we have provided the results from different continents including Europe; where 520.885 subjects were included to the analysis on association of DII and UGI cancers, and 500,223 individuals contributed to analysis of the association of MDS on UGI cancers.

Please see pages 6-8, Table 1; which provides the number of study population and the countries in which the studies have been carried out, and please check supplementary file, figure S2A & Figure S2B which demonstrates the pooled effect size stratified for different continents. Therefore, the required information is provided to have an estimation of the size of population contributing to the pooled findings per geographic region.

2.2. Food availability determines the dietary habits. Diet also has long term effect. Genetics plays a role too. Therefore, considering the complexity, it is not surprising that it is an inconclusive study. I would recommend restricting these meta-analyses to European population, which may suggest areas to focus to obtain more conclusive results.

Reply: We also agree that some degree of inconclusive findings may be due to difference on factors which the reviewer rightfully has referred to and we agree the focus on European population may lead possibly to more conclusive results; we performed stratified analysis by the study populations. We found an effect estimate of OR=0.71 (95%CI: 0.57 to 0.90) for association of DII and of OR=0.72 (95%CI: 0.62 to 0.83) for association of MDS with UGI cancers. These effects are consistent with estimated effect sizes in overall analysis, as presented and please see supplementary file, Figure S2A & Figure S2B. Findings from our study reveals consistent findings with low level of heterogeneity on MDS and UGI cancers in European regions, though for DII and UGI cancers, despite the consistency on observed beneficiary effect, a significant level of heterogeneity exists. Please note that the aim of this study was to provide overview on the quality of pooled findings on diet quality, and UGI cancer risk using all published studies.

2.3. The study included reports published till Feb. 2020. What is the start date for this study? If the study included points from???

Reply: There was not time restriction for the start date. Our search in databases covered all studies included in that database until February 2020.

2.4. The authors mentioned they had evaluated data from 11 studies from other sources. I recommend describing these sources.

Reply: Eleven studies were included via further assessment of the reference list of included studies by searching the database, for a better clarification,  we revised the manuscript on, and please see, page 5, Figure 1.

2.5. I expect this meta-analyses article may be read by a wider audience. Therefore, I recommend providing more details about the databases used. For example, the Cochrane and the Globocan databases have invaluable information.

Reply: The following required information are added to the text on page 3, lines 123-128, page 4, lines 139-141.

“We obtained the required information on cancer incidence rates for the country of corresponding studies, within the follow up period, using the data bases offered by website of Global Cancer Observatory. This web-based platform presents worldwide cancer statistics, with the aim of providing information both for cancer research and cancer control. The data on global epidemiological profile of cancers are supplied by several projects of International Agency for Research on Cancer, including GLOBOCAN, Cancer incidence in five continents and cancer survival; in Africa, Asia the Caribbean and Central America. The required information on applied methodology in this systematic review and pooling the findings, and quality assessment of overall findings were obtained from Cochrane handbook and website [1,20].”

Reviewer 3 Report

A major criticism of the manuscript is that an objective is to determine if dietary recommendations for the prevention of UGI cancer could be developed. Diet is a modifiable factor in reducing risk for selected chronic conditions but not the only factor impacting disease. Thus the idea that diet can prevent such a disease as cancer is incorrect. Given the study designs of research reviewed and the quality ratings determined by the authors, this article adds little to our current knowledge. 

Reference 1 is incomplete

Minor spelling edits - line 81, sase should be case

line245, inconstancy should be inconsistency

Figures S2A, 1.1.9 & S2B,2.1.10 Overal should be Overall

Author Response

Comments from reviewer #3

Regards

Reviewer 4 Report

Abstract:

  • How many studies were included in the analyis for DII and MDS?
  • Did heterogeneity differ between studies looking at DII and MDS?

Methods:

  • The dietary assessment does not appear to be part of the quality assessment? Do the authors believe the the quality of dietary assessment is comparable between studies or is sufficient/good in all studies? The problems of dietary assessments have not been mentioned anywhere in the manuscript. I would at least expect a brief discussion of the difficulties to assess a person's habitual diet in the discussion section. There is only one brief comment in line 271 with references to studies 6 and 12, which, however, is rather confusing because most studies rely on only a single dietary assessment.

Results:

  • It is sometimes not entirely clear if the authors refer to a table/figure in the main results or in the supplementary files. Please check carefully the results section and label correctly.
  • Also, reference to the funnel plots are not correct in the text (lines 138 and 150).

Discussion:

  • Provide references of the meta-analyses mentioned in l. 234 and l. 265.
  • Comparison with other meta-analyses: The authors could provide a bit more information on why the results might differ compared with the current analysis. Eg. why did Due (or Du) et al. include only 3 studies?
  • Conclusion: what is necessary to develop dietary recommendations?

Typos in Supplementary Figures (Asia and Europe are incorrect), which makes it difficult to understand what the authors are showing. It becomes clear once you know what the words actually mean.

References:

Ref. 1 is concomplete!

The meta-analysis by Due et al., mentioned in l. 245, is not cited in the references. Or is it reference 16 (Du S et al.)?

Author Response

Comments from reviewer #4

Regards

Round 2

Reviewer 1 Report

The authors have provided thorough and valid responses to my comments. However, there are several errors (punctuation, font sizes, space) that need to rechecked (e.g., lines 45, 102, number of studies for inclusion/exclusion in figure 1). 

Reviewer 3 Report

Revisions to the manuscript have enhanced its quality.

Author Response

Comment from reviewer #3:

Revisions to the manuscript have enhanced its quality.

Reply: We appreciate the feedback from the respected reviewer.

Reviewer 4 Report

There are still some typos in the text. Eg. Author of reference 16 (Du et al.) is still not spelt correctly in the text (Due instead of Du, line 227).

Supplementary tables A1 and A2 are currently in the folder "Non-published material". However, I assume they will be available online?
